# Different Clinical Features of Idiopathic Achalasia in Various Countries

**Amy Yeung** [1,*] **and Imaan Benmerzouga** [2]

1   West Virginia School of Osteopathic Medicine, Lewisburg, WV 24901, USA
2   Department of Biomedical Sciences, West Virginia School of Osteopathic Medicine, Lewisburg, WV 24901, USA; ibenmerzouga@osteo.wvsom.edu
*   Correspondence: ayeung@osteo.wvsom.edu

**Abstract:** Idiopathic achalasia is a motility disorder affecting the lower esophageal sphincter. Dysphagia is a hallmark symptom, but patients may exhibit other symptoms. The aim of this review is to compare achalasia symptoms globally. PubMed and Google Scholar were filtered from 1952–2021 with the search terms achalasia, epidemiology, diet, countries, and genetics. A total of 14 articles addressed demographics, symptom profiles, genetics, and diagnosis criteria amongst 2463 patients. Data on countries' climate and diet were obtained through Arc Geographic Information System (GIS) and Our World in Data. Countries were grouped by similar climate zones and diets. Achalasia symptoms varied by region. In West Africa, patients exhibit parotid swelling, anemia, and dehydration; diminished appetite in East Asia; dysphagia and weight loss in West Asia and Europe; respiratory symptoms, reflux, and retrosternal pain in North America; and vomiting in Southern Asia. Weighted percentages of dietary oils/fats were (24.3%) in North America, Western Asia (17.8%); Europe (17.7%); East Asia (17.6%); West Africa (14.7%); Southern Asia (13.8%); North Africa (12.4%); Northeast Africa (10.1%). Conditions such as Down Syndrome and Triple A syndrome are associated with achalasia. There was no correlation for achalasia presentation and climate zones. Achalasia symptoms are likely multifactorial. Diet, genetics, and environmental factors may play significant roles.

**Keywords:** achalasia; epidemiology; diet; countries; genetics

## 1. Introduction

Idiopathic achalasia is a rare esophageal motor disorder that mainly affects the lower esophageal sphincter (LES). It is caused by the loss of inhibitory neurons and ganglionic cells in the intrinsic enteric nervous system, particularly the myenteric plexus. This loss of inhibition leads to persistent contraction of the LES [1].

Derived from the Greek translation, "failure to relax", achalasia was first described by Sir Thomas Willis in 1674 as "food blockage in the esophagus" [2]. As treatment, Willis utilized a whale bone and sponge to successfully dilate one's esophagus for food passage [2]. Currently, idiopathic achalasia affects both men and women equally, and the prevalence and annual incidence are estimated to be 1/10,000 and 1/100,000, respectively [1].

Many achalasia patients experience symptoms for years before seeking medical attention. Dysphagia is a classic hallmark symptom and is primarily for solids during onset but may gradually progress to both solids and liquids [3]. A sign of progression of the disease is when patients begin to complain of regurgitation of both solids and liquids. This becomes a problem when left untreated, especially when the esophagus begins to dilate. However, this progression may be difficult to detect, as some patients with achalasia may learn how to accommodate their symptoms, such as by eating more slowly, switching to softer food, or avoiding social events that involve large meals. As a result, achalasia patients may endure these symptoms for many years, which can further delay diagnosis and proper treatment.

There are different imaging modalities on confirming the diagnosis of achalasia. Tests such as esophagogastric-duodenoscopy (EGD) and radiographic barium swallows are used but are not definitive [4,5]. As a result, the gold standard for diagnosis is high-resolution esophageal manometry (HRM), motility testing that consists of a long and flexible catheter with 36 pressure sensors that detect changes throughout the esophagus during one's swallowing process. These pressure changes are detected as color variations in a spatiotemporal scheme. A novel diagnostic modality, EndoFlip, has the ability to measure distensibility levels. Despite these imaging modalities, about 20–50% of achalasia cases are initially misdiagnosed, which can delay appropriate treatment [5].

While there is no definite cure for achalasia, management is often needed and is directed to improve esophageal emptying, prevent further esophageal dilation, and improve patients' quality of life [3,4]. Current management depends on a patient's comorbidities, age, surgical risk, patient preferences, and available mode of treatment [5]. These conventional management methods can include Botulinum toxin injections, pneumatic balloon dilations, esophagectomy, and surgical myotomy [4,5]. Despite these, living with any chronic ailment is difficult, especially for those with idiopathic achalasia. Thus, early recognition of these clinical symptoms is essential to allow for more accurate approaches in the disease course. To provide a better understanding of the role of diet, climate, and genetics in the presentation of achalasia, we performed a literature review comparing the presentation of achalasia in different regions.

## 2. Methods

### 2.1. Selection Process

A literature search was conducted in two databases, PubMed and GoogleScholar, in which articles from 1951 to 2021 were filtered and searched. We applied the eligibility criteria described in Figure 1B and organized the search as suggested by the Preferred Reporting Items for Systematic Reviews and Meta-Analyses (PRISMA) statement [6]. For PubMed, keywords ((achalasia AND (epidemiology OR diet OR food OR diet OR features OR countries OR genetics)) were applied. In GoogleScholar, the keywords were (achalasia AND (epidemiology OR diet OR food OR diet OR features OR countries OR genetics)). A detailed summary of selected articles and the selection process can be located in the Supplementary File S1. A total of 477 articles were obtained, and 12 duplicates were removed using EndNote 20 software. Articles that did not meet the eligibility criteria were excluded from the review. Finally, 14 articles, which included a total of 2463 patients with idiopathic achalasia, were selected for the review. The selection process is shown in the PRISMA flowchart in Figure 1A.

### 2.2. Dietary Composition Data

Data on countries' dietary compositions were extracted from National Geographic and Our World in Data (OWID), which organize specific food commodities into higher-level categories. The higher-level categories include 10 United Nations Food and Agricultural Organization (FAO) items: Cereals and grains, pulses, starchy roots, fruits and vegetables, dairy and eggs, oils and fats, sugar, meat, alcohol, and others. Cereals and grains include wheat, rice, maize, barley, oats, millet, sorghum, rye, and other derivatives. Oils and fats include vegetable oils, animal fats, oil crops, and tree nuts. Meat consists of bovine, poultry, pork, mutton and goat meat, fish, and total seafood. Dairy and eggs do not include butter. Sugar includes sugar crops and sweeteners. The others category includes spices, infant food, and miscellaneous items.

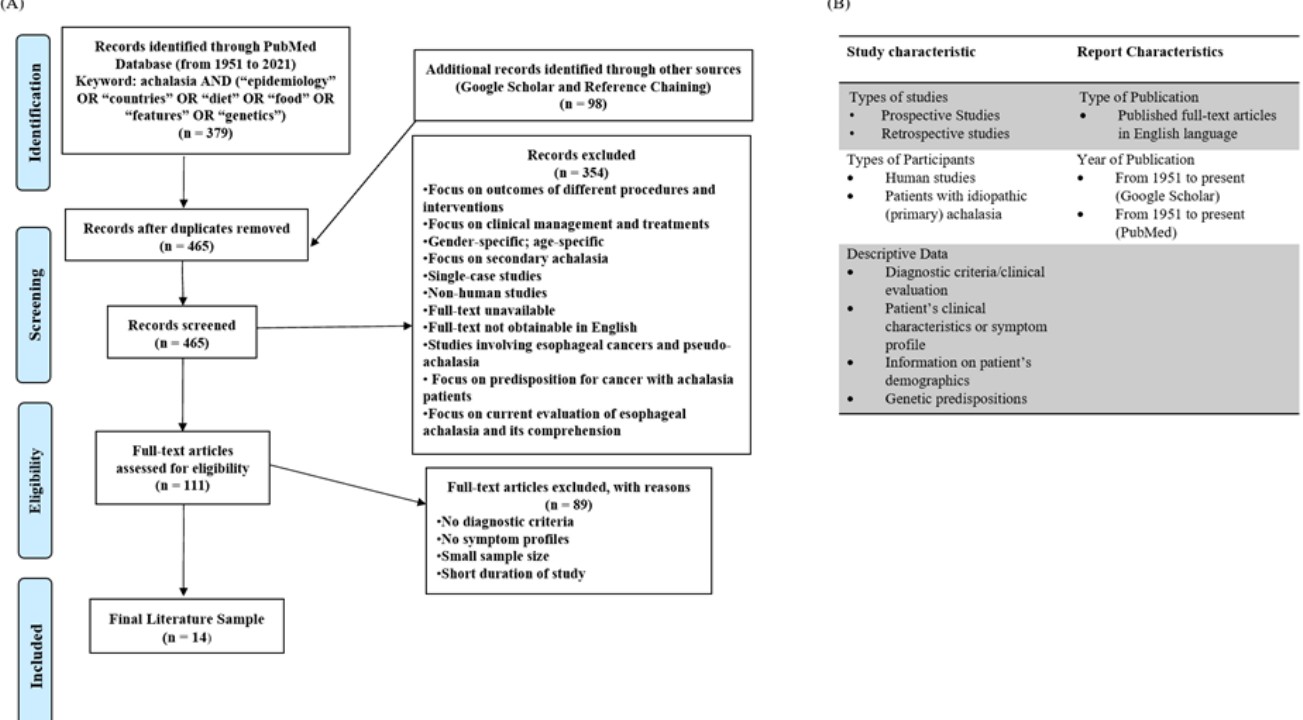

**Figure 1.** (**A**) PRISMA flowchart of the selection process for the studies evaluated in this review. (**B**) Table of eligibility criteria for selecting studies involving idiopathic achalasia and patients' clinical symptoms.

Each country's dietary composition was individually searched and then organized according to the 10 FAO items. Next, the percentages of each food item were derived from each country's total calorie intake. The countries were then grouped by similar regions, in which each region's metrics were weighted by total country population.

## 3. Results

### 3.1. Clinical Characteristics

The prevalent symptoms of achalasia patients in various regions are summarized in Table 1. Based on a retrospective study of 169 achalasia patients conducted in Quebec, Canada, prevailing symptoms were dysphagia (67%), blocking sensation (23%), weight loss (23%), and reflux (16%). Another retrospective study of 165 achalasia patients done in the United States showed dysphagia (100%), regurgitation (97.6%), respiratory symptoms, (41%), and retrosternal pain (77.6%) [7,8].

In West Africa, three studies were conducted on achalasia patients in Nigerian cities, including Ife-Ife, Zaria, and Enugu. In Ife-Ife, 33 achalasia patients showed symptoms of dysphagia (100%), regurgitation (72.7%), >5 kg weight loss (57.6%), postprandial intrathoracic fullness (30%), recurrent pulmonary infections (24.2%), bilateral parotid gland hypertrophy (15.2%), and bradycardia (12.1%). In Zaria, 47 achalasia patients had symptoms of dysphagia (100%), regurgitation (70.2%), heartburn (25.5%), general weight loss (21.3%), parotid swelling (10.6%), and cough (8.5%). In Enugu, 43 achalasia patients showed symptoms of dysphagia (95%), regurgitation (75%), weight loss (52.5%), retrosternal fullness (27.5%), anemia (20%), cough (20%), bilateral parotid sialadenitis (17.5%), and dehydration (15%) [9–11].

**Table 1.** Summarized table of 14 studies and most prevalent symptoms of achalasia in each region.

| Author | Country | Region | Population (N) | Symptoms | Total Percentage (%) |
|---|---|---|---|---|---|
| Pouyez et al. (2019) [7] | Canada | North America | (169) | • Regurgitation<br>• Dysphagia<br>• Retrosternal pain<br>• Respiratory symptoms<br>• Blocking sensation<br>• Weight loss<br>• Reflux | (98)<br>(83)<br>(78)<br>(40)<br>(23)<br>(23)<br>(16) |
| Sinan et al. (2011) [8] | United States | | (165) | | |
| Adeyemo et al. (1987) [9] | Nigeria | West Africa | (33) | • Dysphagia<br>• Regurgitation<br>• Weight loss<br>• Retrosternal fullness<br>• Heartburn<br>• Anemia<br>• Cough<br>• Parotid gland swelling | (98)<br>(72)<br>(42)<br>(28)<br>(26)<br>(20)<br>(14)<br>(14) |
| Ahmed, A et al. (2008) [10] | Nigeria | | (47) | | |
| Ezemba et.al (2007) [11] | Nigeria | | (43) | | |
| Tebaibia et al. (2016) [12] | Algeria | North Africa | (1256) | • Dysphagia<br>• Regurgitation<br>• Weight loss<br>• Heartburn<br>• Respiratory symptoms | (99)<br>(83)<br>(70)<br>(25)<br>(22) |
| Abbas et al. (2016) [13] | Sudan | Northeast Africa | (51) | • Dysphagia<br>• Chest Pain<br>• Weight loss<br>• Heartburn<br>• Wheeze<br>• Cough | (100)<br>(65)<br>(44)<br>(43)<br>(26)<br>(22) |
| Ng et al. (2010) [14] | Hong Kong | East Asia | (32) | • Dysphagia<br>• *Heartburn<br>• Regurgitation<br>• Weight loss<br>• Diminished appetite | (100)<br>*(59)<br>(51)<br>(47)<br>(3) |
| Jeon et.al (2017) [15] | South Korea | | (64) | | |
| Aljebreen et al. (2014) [16] | Saudi Arabia | Western Asia | (29) | • Dysphagia<br>• Regurgitation<br>• Weight loss<br>• Heartburn<br>• Chest pain | (100)<br>(56)<br>(45)<br>(24)<br>(21) |
| Jain et al. (2020) [17] | India | Southern Asia | (452) | • Dysphagia<br>• Regurgitation<br>• Vomiting<br>• Chest pain<br>• Epigastric pain | (94)<br>(80)<br>(54)<br>(35)<br>(30) |
| Ahmed et al. (2008) [18] | Pakistan | | (46) | | |
| Birgisson et al. (2007) [19] | Iceland | Europe | (62) | • Dysphagia<br>• Regurgitation<br>• Chest Pain<br>• Weight loss<br>• Heartburn<br>• Cough | (99)<br>(48)<br>(47)<br>(39)<br>(11)<br>(8) |
| Farrukh et al. (2008) [20] | United Kingdom | | (14) | | |

*Heartburn is seen only in South Korea The total percentage (%) only represents South Korea.

In North Africa, a prospective study in Algeria of 1256 patients showed symptoms of dysphagia (98.9%), regurgitation (83%), retro-sternal, epigastric, or inter-scapular pain (51%), weight loss (70%), heartburn (24.5%), and respiratory symptoms (22.3%) [12].

In Northeast Africa, a prospective study of 51 patients in Sudan showed mainly symptoms of general dysphagia (100%), weight loss (65.21%), chest pain (43.50%), heartburn (43.47%), wheeze (26.09%), and cough (21.74%) [13].

In East Asia, a retrospective study of 32 patients in Hong Kong revealed prevalent symptoms of general dysphagia (100%), vomiting (50%), regurgitation (31%), weight loss (31%), extra-gastrointestinal symptoms (13%), and diminished appetite (3%). In Seoul, Korea, another retrospective study showed a total of 64 achalasia patients with dysphagia (100%), regurgitation (76.6%), chest pain (54.7%), general weight loss (68.7%), and heartburn (59.4%) [14,15].

In Southern Asia, a retrospective study of 452 patients in India showed symptoms of dysphagia for both solids and liquids (94.7%), regurgitation (79.6%), and chest pain (35%). In Pakistan, a study of 46 achalasia patients displayed symptoms of dysphagia (83%), vomiting (54%), and epigastric pain (30%) [16,17].

In Western Asia, 29 patients in Saudi Arabia from a retrospective study exhibited symptoms of dysphagia (100%), daily regurgitation (55.5%), >10 kg weight loss (44.8%), daily heartburn (24.1%), and daily chest pain (20.7%) [18].

In Europe, several studies were conducted in Iceland and England. In Iceland, 62 achalasia patients experienced dysphagia (100%), regurgitation (48%), chest pain (51%), weight loss (38%), heartburn (11%), and cough (8%). In England, 14 achalasia patients mainly exhibited dysphagia (93%), weight loss (43%), chest pain (22%), and vomiting (22%) [19,20].

### 3.2. Dietary Compositions

To evaluate the possible role of dietary composition on the variability of the presentation of achalasia, the 12 countries were grouped into separate regions, and the total weighted consumption of dietary components was calculated. The highest and lowest weight consumption of each commodity food group are shown in Tables 2 and 3. The dietary data of the weight sum consumptions and total calorie intake are shown in Figure 2.

**Table 2.** Regions with the highest weighted consumption of the 10 total commodity food groups.

| Commodity Group | Region with Highest Weight Consumption |
|---|---|
| Oils/Fat | North America (24.39%) |
| Sugar | North America (14.98%) |
| Starchy roots | West Africa (22.81%) |
| Cereals and Grains | Southern Asia (54.2%) |
| Fruits and Vegetables | North Africa (10.29%) |
| Dairy and Eggs | Northeast Africa (11.25%) |
| Meat | Europe (13.94%) |
| Pulses | Southern Asia (5.12%) |
| Alcohol | Europe (4.47%) |
| Others | Western Asia (3.38%) |

**Table 3.** Regions with the lowest weighted consumption of the 10 total commodity food groups.

| Commodity Group | Region with Lowest Weight Consumption |
|---|---|
| Oils/Fat | Northeast Africa (10.10%) |
| Sugar | West Africa (3.89%) |
| Starchy roots | East Asia (1.18%) |
| Cereals and Grains | North America (22.19%) |
| Fruits and Vegetables | West Africa (4.41%) |
| Dairy and Eggs | West Africa (0.89%) |
| Meat | Southern Asia (1.29%) |
| Pulses | Southern Asia (5.12%) |
| Alcohol | Western Asia (0.00%) |
| Others | Western Africa (0.52%) |

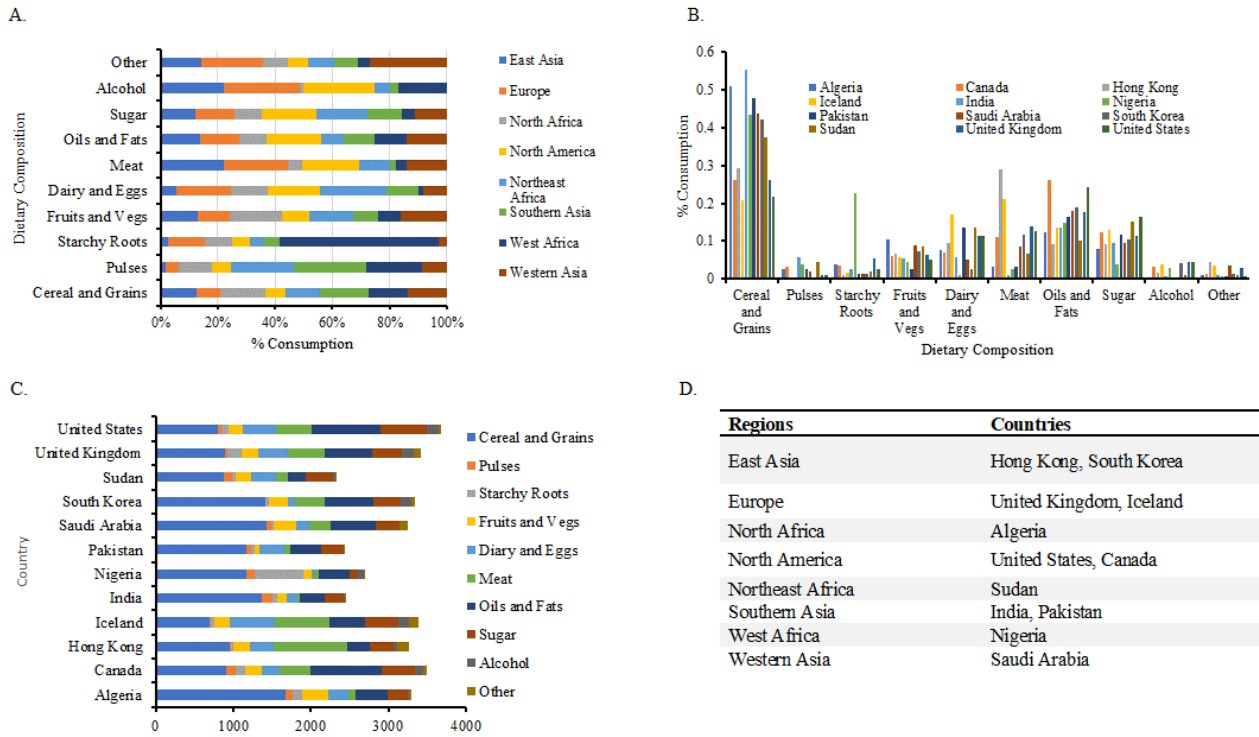

**Figure 2.** (**A**) Weighted sum percentages of dietary compositions based on population density in different regions. (**B**) Weighted sum percentages of dietary compositions based on population density in different countries. A noticeable pattern is the highest consumption of meat shown in Hong Kong (29%) and lowest consumption of meat in India (0.9%). (**C**) Total calorie consumption of dietary composition based on different countries. United States has the greatest number of calories consumed (3682 calories), followed by Canada (3494 calories), United Kingdom (3424 calories), Iceland (3380 calories), South Korea (3334 calories), Algeria (3296 calories), Hong Kong (3260 calories), Saudi Arabia (3255 calories), Nigeria (2700 calories), India (2459 calories), Pakistan (2440 calories), and Sudan (2336 calories). (**D**) Summary of all regions and the associated countries.

### 3.3. Genetics

There are certain candidate genes associated with idiopathic achalasia. The most prominent ones are polymorphisms in Nitric oxide synthase (NOS), Vasoactive Intestinal polypeptide (VIPR1) gene, Interleukin-23 receptor (IL-23R) gene, and protein tyrosine phosphatase non-receptor 22 (PTPN22), which all have been shown to potentially increase one's risk of achalasia [1,21].

Nitric Oxide Synthase (NOS) is a widely studied candidate gene that is thought to mediate esophageal motor function. It utilizes L-arginine to synthesize nitric oxide, an inhibitory neurotransmitter in the esophageal myenteric plexus [22]. In a normal esophagus, nitric oxide allows for relaxation and proper food bolus propulsion in the esophagus [3]. There are three isozymes of NOS: neuronal NOS (nNOS), inducible NOS (iNOS), and endothelial NOS (eNOS). nNOS is known to be the main source of NO in the LES and is suggestively involved in gastrointestinal abnormalities seen in achalasia [22].

Vasoactive intestinal peptide is also an inhibitory neurotransmitter that plays a similar role as Nitric Oxide synthase based on its involvement in smooth muscle relaxation in the esophageal myenteric plexus. In addition, it may also play another role in regulating inflammation [1].

Another selected candidate gene is protein tyrosine phosphatase non-receptor 22 (PTPN22), a type of phosphatase that controls T-cell activation and may be associated with

several autoimmune diseases, such as Rheumatoid arthritis and Type I Diabetes [1]. Other candidate genes are described in Table 4.

**Table 4.** Description of prominent candidate genes that may be increase one's susceptibly to idiopathic achalasia.

| Candidate Genes | Function | Isozymes | Chromosome Location |
|---|---|---|---|
| Nitric Oxide Synthase [21] | Synthesizes nitric oxide | Neuronal NOS (NOS1)<br>Inducible NOS (NOS2)<br>Endothelial NOS (NOS3) | chromosome 12q24.2<br>chromosome 17q11.2-q12<br>chromosome 7q36 |
| Vasoactive intestinal polypeptide (VIPR1) gene [21] | A small neuropeptide that acts as a neurotransmitter with anti-inflammatory properties. This is found in the myenteric plexus to regulate relaxation of the LES and distal esophagus | | chromosome 3p22 |
| Interleukin-23 Receptor (IL-23R) gene [23] | A Type I transmembrane protein that is expressed by Th17 cells. These are associated with inflammatory and chronic autoimmune disorders | | chromosome 1p31 |

## 4. Discussion

### 4.1. Clinical Features and Diet

Achalasia is a primary esophageal motility disorder that presents with dysphagia as a distinct symptom. Although it is known that patients may complain of other symptoms, our study analyzes the variation of the presentation of achalasia symptoms on a global scale. Interestingly, we found that the frequency of heartburn was highest in East Asia (59.4%), which comprises Hong Kong and South Korea. Achalasia patients in South Korea showed a (59.4%) heartburn frequency. This observation is likely due to South Korea's rapid nutrition transition, in which intake of meat and poultry increased 10-fold between 1969–1995 and the intake of milk and dairy products increased 4.3-fold between 1980–1985 [24]. Although our data show that Hong Kong has a higher meat intake (28.8%) and dairy consumption (9.0%) than those in South Korea, (11.6%) and (2.5%), respectively, we believe that this sudden introduction of previously non-traditional food items to South Korea may have contributed to heartburn.

Second, our data show weight loss in all the regions, except for achalasia patients in the Southern Asia region, which consists of Pakistan and India. Perhaps this variability is due to the high intake of calorie-dense carbohydrates such as pulses and cereal/grains, seen mainly in India (55.3%) and Pakistan (48%), as shown in Figure 2C and Table 2. It is possible that early diagnosis and intervention in these countries could potentially impair the classical presentation of weight loss seen in patients with achalasia. Although our data show that India and Pakistan's total calorie intake are lower compared to other countries, it is also possible that they have already changed their dietary intake or used various self-taught accommodating maneuvers to relieve their dysphagia and regurgitation symptoms, which may have not influenced weight loss. However, it is still unclear why patients report weight loss compared to others, and this needs to be further investigated.

Interestingly, it seems that patients with achalasia in Nigeria were the only ones who experienced parotid gland swelling as a symptom. This could be possibly due to the high intake of starchy roots (22.81%), as shown in Figure 2C and Table 2, which makes up most of their total calorie intake. In addition, Nigeria has the lowest consumption of dairy and eggs (0.89%), fruits and vegetables (4.41%), and sugar (3.89%) compared to other countries. Based on our data, Nigeria has the fourth lowest total calorie consumption of 2700 calories.

This raises the possibility that the quantity of consumption of certain food groups could be responsible for this symptom.

Fourth, we notice that North America has the lowest prevalence of weight loss compared to all the regions. This may be theorized by their high intake of oils and fats (24.39%), which is shown in Figure 2C and Table 2. This can also be possibly explained by North America having the highest total calorie consumption amongst the other regions. On the other hand, we observe that North Africa has the highest percentage of weight loss (70%), which may be explained by its high intake of fruits and vegetables (10.29%) compared to the other regions.

To our knowledge, there are currently no published studies evaluating nutrition and dietary intake in achalasia patients in different regions. Based on these observations and patterns, diet must have an important role in certain achalasia symptoms. While various management methods may help reduce patient symptoms, focusing on a patient's diet and lifestyle might be more significant.

### 4.2. Climate

Data were extracted from Arc Geographic Information System (GIS) database, and each country was grouped together based on similar climate. Given the scarce evidence to support an environmental etiology and its role in achalasia symptoms, we were unable to recognize any patterns of its association. Therefore, future studies are warranted to further investigate climate's role in achalasia symptoms.

### 4.3. Genetics

Our study reviewed several candidate genes and their polymorphisms, such as Nitric Oxide Synthase, Vasoactive intestinal polypeptide gene, and Interleukin 23- Receptor gene. However, these studies only emphasized one's susceptibility to and increased risk of achalasia. Thus, the authors failed to explain the genetic deposition and its role in the development of certain achalasia symptoms.

We also reviewed current knowledge of the established associations between achalasia and isolated well-known diseases, including Down syndrome, (MEN2)B syndrome, Triple A syndrome, congenital central hypoventilation syndrome, Smith-Lemli-Optiz syndrome, and Riley-Day syndrome [21,22]. However, these conditions and their defined disease-causing genes do not fully explain the observed variability in symptoms among patients with achalasia.

Nevertheless, the association of genetics and diet are not well recognized. While there is a wide genetic diversity among humans, we still do not know whether these candidate genes are mainly responsible for idiopathic achalasia and their possible influence on diet. Thus, understanding the mechanism and role of genetics in achalasia is important for the future development of new treatment possibilities.

### 4.4. Limitations/Strengths

We acknowledge several limitations in our study. First, there are various sample sizes, as the studies were conducted at a single hospital or a tertiary center. These sample sizes may be a great challenge for future studies of idiopathic achalasia and may warrant a multi-center approach. Second, we are also aware that several countries used different diagnostic approaches, including interpretation of patient's clinical symptoms or utilizing different modes of imaging as gold standard tests. Third, our study only disclosed patients' regional area and not their ethnicity or origin, which could be a possible factor in certain symptoms. Future study of this area may be needed to provide further information of this factor. Fourth, there are no reported histories of the patients' dietary habits; the reported dietary compositions reflect only the general population of a country.

We also readily acknowledge several strengths, as we use objective statistical analysis of the reviewed countries to find mendable causes of achalasia. In addition, we also obtain both quantitative and qualitative data from reliable sources on total dietary composition.

Third, we highlight various and exclusive achalasia symptoms in addition to the most common symptoms. Lastly, our study consists of selected population groups in each country that met certain eligibility criteria, which minimizes selection bias.

## 5. Conclusions

Idiopathic achalasia is a rare and chronic esophageal motility disorder that is characterized by the failed relaxation of the LES. Patients with achalasia often complain of dysphagia but may exhibit other clinical symptoms. Esophageal manometry is the gold standard diagnostic test, but early diagnosis can be difficult to capture, as achalasia patients may learn how to accommodate for their symptoms and delay medical attention. This could result in a gradual and debilitating progression of their disease. Current surgical, endoscopic, and pharmaceutical interventions are used to help relieve patient symptoms but are not definitively curative [4,25].

In this review, an overall summary of symptom presentation in relation to the country of diagnosis is presented. Aside from dysphagia being the cardinal symptom regardless of country, the other clinically relevant presentations varied between countries. Interestingly, when dietary compositions were analyzed based on region and country, several observations were made.

First, the frequency of heartburn is seen as highest in South Korea, which may be possibly due to its rapid non-traditional food transition to a more meat-and-dairy diet. Second, weight loss is not seen in India and Pakistan, which may be explained by their high calorie intake of pulses, cereals, and grains. This could suggest a role for diet in the presentation of clinically important symptoms of achalasia. Third, North America has the lowest prevalence of weight loss, which may be due to its high intake of oils and fats. Fourth, parotid gland swelling is only seen in achalasia patients in Nigeria, which is possibly associated with their high intake of starchy roots. Other variables such as climate, candidate genes, and associated conditions were also reviewed for this study but there were no significant observations made.

These findings in our study highlight that diet may influence the symptoms of achalasia; however, further investigation requires a more solid reasoning. With regard to patient management, we are hopeful that such investigation will address unmet needs and guide more tailored clinical practices and future adjunctive treatments. Future dietary modification and behavioral changes may be promising and aid patients with achalasia to have better health outcomes and experience more sustainable and stress-free lives.

**Supplementary Materials:** The following are available online at https://www.mdpi.com/article/10.3390/gidisord4020007/s1.

**Author Contributions:** Conceptualization, A.Y.; Methodology, A.Y., I.B.; Investigation, A.Y.; Writing—Original Draft Preparation, A.Y.; Writing—Review and Editing, I.B. and A.Y.; Supervision, I.B.; Project administration, I.B. All authors have read and agreed to the published version of the manuscript.

**Funding:** This research received no external funding.

**Acknowledgments:** We like to acknowledge the West Virginia School of Osteopathic Medicine for providing the resources to support this completion of this project.

**Conflicts of Interest:** The authors declare no conflict of interest.

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
