# Peer review of "Different Clinical Features of Idiopathic Achalasia in Various Countries"

_gastrointestdisord, doi:10.3390/gidisord4020007_

Round 1

Reviewer 1 Report

The article is good, the idea is interesting, it is well planned and well presented, the maths are good and the results are reliable. On the other hand the table regarding the persentage of every single symptom is confusing. I would chose a scale going from the highest to the lowest percentage or a fixed classification of the symptoms (so the percentage would refer to the symptom independently of its position on the scale).

Like:

Dysphagia (83)                                               Regurgitation 98

Regurgitation (98)                                           Dysphagia 83

Respiratory symptoms (40)                              Retrosternal 78

Weight loss (23)                                               Respiratory symptoms 40

Reflux(16)                                                         Weight loss 23

Retrosternal pain (78)                                       Blocking sensation 23

Blocking sensation(23)                                     Reflux 16

Reviewer 2 Report

In this review the authors assessed different clinical features of patients with idiopathic achalasia.

Major suggestions

It is useful to explain before the section Results (not at the end, at least that the format of the journal requires this style), the method used to choice the  papers discussed in this narrative review (key words, source) .

Minor suggestions:

Line 73, it is known that North America comprises US and Canada. Hence, this sentence should be removed or modified if the authors believe that it is important for some reason. Similarly, for the sentence line 89 "in North Africa which comprises Algeria"

Section Results: at the end of each sentence the appropriate reference should be reported 

At the begin of the section discussion the point 3.1 should be "Clinical Features and Diet", so the authors could highlight /as already done) on differences and similarities in clinical presentation found in the different areas

Line 307, not only medical and surgical approaches are proposed but also endoscopic (see for example the update by Khoury et al. Minerva Gastroenterol 2021;67(2):171-2). Please add it.

Line 324-325, this sentence should be modified, since there are no evidence to support that diet could ba a cause of achalasia. It could be reported that diet could influence the diagnosis or that achalasia influemce the diet.

Due to the evolution of knowledge in this field references should be updated. I suggest to change the number 4 (2016) with Mari A et al. Diagnosis and management of achalasia: updated of the last two years. J Clin Med 2021;10(16):3607. 
